# Spillover of Vaccine Hesitancy into Adult COVID-19 and Influenza: The Role of Race, Religion, and Political Affiliation in the United States

**DOI:** 10.3390/ijerph20043376

**Published:** 2023-02-15

**Authors:** Victoria Zhang, Peiyao Zhu, Abram L. Wagner

**Affiliations:** Department of Epidemiology, School of Public Health, University of Michigan, 1415 Washington Heights, Ann Arbor, MI 48109, USA

**Keywords:** COVID-19 vaccines, survey and questionnaires, demography

## Abstract

Background: Due to its potential to lead to vaccine delays and refusals, vaccine hesitancy has attracted increased attention throughout the COVID-19 pandemic. It is crucial to investigate whether demographic patterns differ between adult general vaccine hesitancy and COVID-19 and flu vaccine non-receipt. Methods: A cross-sectional survey was conducted online in August 2022. In response to questions about vaccine hesitancy, participants indicated whether they would receive the vaccine given various safety and efficacy profiles. Through logistic regression models, we examined variations between general vaccine hesitancy and COVID-19 non-vaccination. Results: Among the 700 participants, 49% of the respondents were classified as having general vaccine hesitancy, 17% had not received the COVID-19 vaccine, and 36% had not had flu vaccinations. In the multivariable analysis, general vaccine hesitancy and the non-receipt of COVID-19 vaccines were significantly higher in Non-Hispanic Black participants, those with no religious affiliation, and Republicans and Independents. Conclusions: Patterns of vaccine hesitancy and the non-receipt of the COVID-19 vaccination did not vary, indicating a substantial overlap and potential spillover in vaccine hesitancy over the course of the pandemic. Because changing people’s opinions regarding vaccinations is generally a challenge, different interventions specific to demographic subgroups may be necessary.

## 1. Introduction

The COVID-19 pandemic is one of the greatest challenges during the 21st century, and it could reshape the landscape of public health, including how the general population responds to governmental recommendations [1]. Since the COVID-19 pandemic’s initial emergence in December 2019, it has become one of the biggest public health crises in the last century [2]. Through December 2022, it has caused more than 6.6 million deaths worldwide [3]. The public health responses include the invention and implementation of safe and effective vaccines and other non-pharmaceutical interventions such as social distancing and mask-wearing [4]. Many efforts have been made to create vaccines against COVID-19 [5]. 

Even before the COVID-19 pandemic, vaccines were considered one of the ten great public health achievements in the United States in the 20th century in eliminating or controlling various infectious diseases [6]. They bring direct protection for immunized individuals, and extensive immunization, in some cases, can lead to indirect shielding for the entire community due to the decreased transmission of vaccine-preventable disease (VPD), creating a herd immunity that lowers the risk of infection among vulnerable people in the community [7]. Hesitancy in obtaining vaccines has long been recognized as a threat to individual and population protection against VPDs [8]. 

In March 2012, the World Health Organization Strategic Advisory Group of Experts (SAGE) on Immunization created a Working Group on vaccine hesitancy which defined vaccine hesitancy as “a delay in acceptance or refusal of vaccination despite availability of vaccination services. Vaccine hesitancy is complex and context specific, varying across time, place, and vaccines. It is influenced by factors such as complacency, convenience and confidence” [7]. This definition acknowledges a difference between vaccine hesitancy and the non-receipt of vaccines; individuals who have received vaccines would still have mistrust and concerns about some aspects of vaccines [9,10]. Additionally, hesitancy towards one vaccine may or may not correlate with hesitancy towards another vaccine.

Vaccine hesitancy has historically been more well studied among parents in reference to childhood vaccines [11,12,13]. Previous studies have demonstrated that vaccine hesitancy is a global problem [14,15,16] and can affect attitudes towards a range of vaccines, such as the influenza and measles/Mumps/Rubella (MMR) vaccines [17]. For example, in the US, where the measles virus was formally declared eliminated in 2000, there have been declining MMR vaccination rates in particular communities (e.g., such as ultra-Orthodox Jews in New York, where a measles outbreak persisted in 2019) [18,19]. Imported measles cases into the US can lead to secondary cases and outbreaks, especially among unvaccinated persons [20,21]. Similar phenomena in other nations caused a 30% rise in measles infections globally [18]. As another example, the influenza vaccine uptake in different areas is suboptimal. Researchers found declines in influenza vaccination coverage among Hong Kong nurses in 2017–2018, with 79% of them expressing moderate or greater vaccine hesitancy [22]. From a cross-sectional survey in Canada, from 2006/07 to 2013/14, only 29% of respondents said that they had received a seasonal influenza vaccine in the past year, compared to a target 80% vaccination rate for high-risk groups [23].

The attitude towards vaccines depends on the particular vaccine and can vary by group, including race/ethnicity, political affiliation, and religion. Studies have shown that Non-Hispanic (NH) Black, Hispanic, and American Indian/Alaskan Native people have relatively low influenza vaccination rates [24]. Previous studies have linked religious beliefs to vaccination decisions [25,26,27]. Moreover, there has been a rise in the frequency of religious exemptions for vaccines [28]. In a specific example, researchers have discovered that, in the 2007–2009 mumps pandemic in the Netherlands, most cases were unvaccinated children from conservative Protestant families [28]. 

The roll-out of COVID-19 vaccines has highlighted the need to comprehensively understand adult vaccine hesitancy and its manifestation within the population. Hispanic and Black Americans have been disproportionately affected during the pandemic, resulting in poor health outcomes [29,30,31]. Yet, generalized mistrust of the health system has led to substantial vaccine hesitancy and the non-receipt of COVID-19 vaccines among the Black community [30]. According to a 2020 Kaiser Family Foundation (KFF) survey, almost half (48%) of Black adults said they are not confident that the COVID-19 vaccine is being developed with Black people’s needs in mind, and 35% of Black adults claimed they definitely or probably would not get vaccinated [31].

Political affiliation has also been shown to play a significant role in vaccine decision making. This relationship may be due to shifts in the social norms regarding vaccines among right-wing communities and news media [32,33]. A KFF survey showed that political affiliation was a stronger predictor at the start of the pandemic in terms of mask wearing, social distancing, and receiving vaccines. By October 2021, a large majority of unvaccinated US adults identified as Republican or Republican-leaning (60%), with only 17% identifying as Democrat or Democrat-leaning [34].

Religion could map onto the prior two groups, or it could independently reflect social norms, mistrust in authorities, and naturalistic beliefs [35]. This could lead to mistrust in vaccines, the decline in medical recommendations, and, thus, increased hesitancy towards vaccinations. Given the important role that religion plays in the US [36], it is important to analyze the impacts of religion on vaccine hesitancy and vaccine non-receipt.

Vaccine hesitancy has been complicated by the COVID-19 pandemic, and attitudes about COVID-19 vaccination could be spilling over into other adult vaccines. Depressed vaccination coverage in certain groups could lead to the persistence of barriers in the burden of infectious disease or solidify norms against vaccination. Politics, race, and religion are important aspects of one’s identify in the US, and we need to better understand how these have mapped onto a range of vaccination outcomes. The specific objectives of this paper are to assess (1) the prevalence of vaccine hesitancy and COVID-19 and flu vaccine non-receipt in the US, (2) the demographic features related to vaccine hesitancy and the non-receipt of adult vaccines, and (3) whether the patterns of vaccine hesitancy and the non-receipt of adult vaccines differ.

## 2. Material and Methods

### 2.1. Study Population

A cross-sectional survey was conducted during August 2022 in the United States by the survey research firm Dynata. The eligible population included residents of the US who were at least 18 years old. This was a convenience sample, and participants were recruited via social media and advertisements. The data collection process occurred with age/gender quotas—oversampling those <45 years of age. Using the US 2020 Census as a guidepost, weights were created based on age, gender, region of country, education, and race/ethnicity to make the survey population generalizable to the US as a whole. We omitted individuals who took less than 180 s to complete the survey, which we judged to be the shortest time required to thoughtfully consider each question, based on extensive pretesting from multiple individuals.

Our sample size calculation was based on the overall project’s goal to precisely estimate the proportion of people who were vaccinated at a particular period, with a margin of error of 4%, an alpha of 0.05, and a power of 80% [37]. A sample size of 800 would be required to estimate an outcome proportion of 50% (a statistically conservative estimate of the target outcome).

### 2.2. Measurement of Vaccination

We used the adult Vaccine Hesitancy Scale (aVHS) to assess vaccine hesitancy [38]. Briefly, this is a 10-item scale, and the sum of each question’s responses is dichotomized at 25 to partition individuals into being hesitant or not. The face validity, concurrent validity, and reliability of this question are reported elsewhere [38].

For COVID-19 vaccination, participants were asked at the start of the questionnaire: “Have you personally received at least one dose of a COVID-19 vaccine or not?” Those who answered “No” are categorized as having not received the COVID-19 vaccine. Vaccine questions were based on survey items from the Kaiser Family Foundation COVID-19 Vaccine Monitor.

The non-receipt of the flu vaccine was obtained by asking participants: “Have you received a flu shot for the current flu season that began October 2021 or not?” Those who answered “No” are categorized as non-receipt of the flu vaccine.

### 2.3. Measurement of Sociodemographic Status

Participants were divided into NH white, NH Black, Hispanic, and other groups based on race/ethnicity questions from the US Census. Participants were questioned regarding their racial background (with multiple options available) and ethnicity (Hispanic, Latino, or of Spanish origin). Those selecting multiple choices were subsequently asked which groups best represented their races. Religion was asked about following the Pew Research Center protocol [39], and respondents were asked about their religion, which was divided into Catholic/Orthodox, Evangelical, Jewish, Mainline Protestant, Muslim, Other Christian, and Not Religious. Buddhism, Hinduism, Islam, and Judaism are all included under “Other religions”. Again, using the Pew Research Center standards, political affiliation was asked about with the question, “In politics today, do you consider yourself a Republican, Democrat, or Independent?”

Age, education, and region were the other covariates considered in the model. We asked people about their gender identity using the categories Female, Male, and Other, as suggested by the American Association of Public Opinion Researchers [40].

### 2.4. Statistical Analysis

A baseline descriptive statistics analysis was conducted in order to determine the demographic characteristics of the sample, as well as the prevalence of general vaccine hesitancy, the non-receipt of COVID-19 vaccines, and the non-receipt of flu vaccines according to each demographic characteristic. The prevalence of receiving both flu and COVID-19 vaccinations, either one of them, or neither of them in the population was also analyzed. Categorical variables are expressed as counts ± percentages. A chi-square test was used to determine the *p*-value, except for variables with low cell counts, where Fisher’s exact test was used to adjust for a small sample size.

Multivariable logistic regression was run to assess whether these demographic variables affect vaccine hesitancy and the non-receipt of COVID-19 vaccines. Vaccine hesitancy and the non-receipt of COVID-19 vaccines were modeled separately, with each individual variable included in the model and controlling for other remaining variables. All analyses were performed after weighing the samples. Odds ratios (OR) and 95% confidence intervals (CIs) are presented from the regression models.

To assess whether there are different patterns in vaccine hesitancy and the non-receipt of COVID-19 vaccines by demographic group, the two outcome variables were extracted and combined as a single dichotomous variable, with one representing both vaccine-hesitant and COVID-19 vaccine-non-receipt people. Multivariate logistic regression was run, including this hesitancy-non-receipt variable and the demographic group interaction terms. The interaction between hesitancy-non-receipt and the demographic group indicates whether there were significant differences in the patterns of vaccine non-receipt and vaccine hesitancy between sociodemographic groups. Because we controlled for race, religion, and political affiliation simultaneously, we interpret each value as a direct effect of the specific sociodemographic status on the vaccination outcome. 

Statistical significance was calculated based on joint tests with a threshold of *p* < 0.05. All statistical analyses were conducted by SAS version 9.4 (SAS Institute, Cary, NC, USA).

## 3. Results

The total number of responses was 806; with consent granted, the number decreased to 751. Omitting individuals who took less than 180 s left a sample size of 700 (87% of all recorded responses).

In Table 1, the patterns of sociodemographic variables are listed for the outcomes: vaccine hesitancy, non-receipt of COVID-19 vaccine, and non-receipt of flu vaccine. A large proportion of the respondents were NH white (65%), Democratic (40%), non-religious (29%), or evangelical Christian (25%). Around a quarter (28%) of the respondents had a Bachelor’s degree (28%). Most of the participants (52%) were older than 45. There were about as many men (48%) as there were women (52%), and slightly more people came from the South (36%).

### 3.1. Overall Vaccination Characteristics

We categorized 41% of the sample as vaccine hesitant, 17% had not received any dose of COVID-19 vaccination, and 36% had not received a flu vaccine in the previous year (Table 1). Race/ethnicity, religion, and political affiliation were significantly related to all three of these outcomes. 

Vaccine hesitancy was relatively high among NH Black (60%) and Hispanic (48%) participants compared to that among NH white (35%) Americans. Conversely, the non-receipt of COVID-19 and flu vaccinations was relatively low among Hispanic (10% and 32%, respectively) Americans, while it was high among NH Black Americans (39% and 59%, respectively). 

By religion, vaccine hesitancy was relatively high among Evangelical Christians (46%) and those with no religious affiliation (42%), and it was low among Catholic/Orthodox (32%), Mainline Protestant (27%), and Muslim (34%) groups. The non-receipt of COVID-19 and flu vaccines was relatively high among those unaffiliated (28%, and 27%), with non-receipt being similar between Evangelical and Catholic/Orthodox participants, and only slightly higher than that among Mainline Protestant groups. 

Vaccine hesitancy was relatively high among Independents (45%) and Republicans (47%), with Independents having higher proportions of not having received a vaccine (27% not receiving a COVID-19 vaccine; 51% not receiving a flu vaccine).

### 3.2. COVID-19 and Flu Vaccination Non-Receipt

In total, 61% (488) of the respondents had both the COVID-19 and flu vaccinations, 22% (176) only received the COVID-19 vaccination, 3% had only the flu vaccination, and 14% had neither (Table 2). Race, religion, and politics all showed significant trends (*p* < 0.0001, *p* < 0.0001, and *p* = 0.004). 

Individuals who were Hispanic (66%) were more likely than NH white (65%) or NH Black participants to receive both vaccinations, while NH Black participants were least likely to obtain both (36%). The percentage of those receiving both vaccines was the highest among Muslims (93%) and Republicans (70%) and the lowest among the Other (37%) and Independents (47%) groups. 

The rate of individuals who had only received the COVID-19 vaccination was the highest among NH Black (25%) and Hispanic participants (25%), followed by Other Christians (39%) and Independents (27%), and it was the lowest among NH white participants (19%), Muslims (7%), and Republicans (14%).

The rates of only receiving the flu vaccine were generally low across all groups, with NH Black people (5%), and those with no religious affiliation (7%) having the highest proportion. The proportion of flu vaccinations among the three political parties is comparable (3%).

As for those who received neither vaccine, the percentage was the highest in the NH Black (35%), Other (31%), and Independent (24%) groups, and it was the lowest among Hispanic (7%), Muslim (0%), and Democrat (7%) participants.

### 3.3. Vaccine Hesitancy

In the model adjusted for all of the shown variables (Table 3), there were also significant differences by race, religion, and politics. The odds of vaccine hesitancy were 3.0 times higher among NH Black participants compared to those among their NH white counterparts (95% CI: 1.5, 6.0). Similarly, people with Other as their religion had odds that were 2.8 times higher than those of those who were Catholic or Orthodox (95% CI: 1.3, 6.1). Republicans had odds that were 2.3 times greater than those of Democrats (95% CI: 1.3, 4.1). 

### 3.4. Non-Receipt of COVID-19 Vaccination

Significant racial and political differences in COVID-19 non-vaccination were also seen in the model adjusted for all of the shown variables (Table 3). Individuals who were NH Black had 3.7-times-higher odds of not receiving the COVID-19 immunization than participants who were NH white (1.8, 7.8). Republicans had odds that were 2.6 times higher than those of Democrats (1.2, 5.5). There were no significant results in the multivariable analysis for the non-receipt of COVID-19 vaccines in either religious group.

In a model encompassing both vaccine hesitancy and the non-receipt of the COVID-19 vaccine, we did not find significant differences in the strengths of the association for these outcomes within each demographic group (see the *p*-values in Table 3). This indicates that the patterns for the non-receipt of a COVID-19 vaccine and for vaccine hesitancy were similar.

## 4. Discussion

The study examined how race, religion, and political affiliation impacted general vaccine hesitancy and COVID-19 and flu vaccine non-receipt in the US during the pandemic. We surveyed 700 participants in August 2022. We found significant patterns in the non-receipt of the flu and COVID-19 vaccines by race, religion, and political affiliation.

A large focus of this analysis was to discover if demographic patterns varied between vaccine hesitancy, COVID-19 vaccine non-receipt, and flu vaccine non-receipt. We did not find any substantial, significant difference in patterns of vaccine hesitancy and COVID-19 non-vaccination. This finding also speaks to the strong possibility of the continued spillover in attitudes across vaccines. The definition of the WHO SAGE Working Group on Vaccine Hesitancy stresses that vaccine hesitancy is type-specific, i.e., could vary across vaccines for different pathogens [8]. However, our study, although finding that the *prevalence* of the uptake for various vaccines may vary, found that the *patterns* of uptake or vaccine hesitancy are largely the same between different sociodemographic groups. As a result, surveys of vaccine hesitancy in general could identify hot spots of non-vaccination for this outbreak and others. Additionally, these trends speak to the difficulty of increasing the uptake of COVID-19 vaccination, as changing minds about vaccines in general is relatively difficult [41], and the interventions needed to do so would be different than increasing the access to or convenience of vaccination. 

Vaccine hesitancy was relatively high among NH Black participants, those with no or an “Other” religious affiliation, and Republicans. There was a substantial, but not exact, overlap between vaccine hesitancy and the non-receipt of the COVID-19 and flu vaccines. These findings are consistent with previous studies, such as when Sharma et al. found out that Republican political identification and vaccine hesitancy were significantly correlated [42] and when Cowan et al. concluded that, instead of only personal characteristics, political identity could account for the growing partisan divide in vaccination coverage [43].

The only variation in our study is that Republicans had the highest rate of receiving both the COVID-19 and flu vaccinations (70%), followed by Democrats (65%). Age-related confounding effects are one explanation that could apply. Since older participants tend to get immunized and Republicans are more likely to be older [44,45], age may explain some of this association, although we note that, in our multivariable model that controlled for age, Republican affiliation was associated with both higher vaccine hesitancy and higher odds of the non-receipt of COVID-19 vaccination. Those Republicans who were hesitant to receive the vaccines but complied are the “hesitant compliers” [9,10]. For instance, they may have concerns about whether a specific, recently introduced vaccination lowers the risk of infection while still believing that tested vaccines are generally effective. The hesitant attitudes could be explained by right-leaning media coverage of the virus and, thus, different partisan responses regarding vaccines [32,33]. During the start of the pandemic, denial and misinformation about the coronavirus erupted on right-leaning media sources. For instance, Fox News, more popular among Republican voters, was much more likely than CNN or MSNBC to use a skeptical narrative that minimized the dangers posed by the COVID-19 outbreak, including terms like “normal flu” and “political weapon” [46]. Only 38% of Fox News viewers expressed concern about the coronavirus in a March 2020 survey, compared to 71% of CNN viewers [47].

Significantly different patterns of vaccine hesitancy and the non-receipt of COVID-19 vaccines were seen by race. Vaccine hesitancy and the non-receipt of the COVID-19 vaccine were relatively high among NH Black participants. Similar results were found in other studies. For instance, Black people were 3.84 times more likely to report being unsure or unwilling to undergo COVID-19 vaccination than their white counterparts [48]. KFF studies found out that, in 2020, 42% of Black Americans reported that they would definitely or probably get vaccinated [31]. Although, by March 2021, these percentages significantly increased, with 61% of Black Americans reporting that they would be willing to get vaccinated or were already vaccinated. The disparity in receiving the COVID-19 vaccine persisted in Black American populations due to racial inequalities and mistrust of the government [31]. The historical roots of racism and segregation have resulted in a racial disparity in COVID-19 diagnoses, hospitalizations, and mortality. For instance, Black Americans had a 1.7-times-higher mortality rate, a 2.2-times-higher risk of being hospitalized, and a 1.1-times-higher chance of receiving a COVID-19 diagnosis than NH white Americans as of November 2022 [49]. This COVID-19 disparity further exacerbated the mistrust and vaccine hesitancy; some researchers found out that, by race/ethnicity, there were substantial differences in healthcare provider trust, and if NH Black Detroiters had levels of healthcare provider confidence comparable to those of NH white Detroiters, 23% of the differences in vaccination uptake by race could be removed [50].

As for the significant differences between religions in vaccine hesitancy and the non-receipt of COVID-19 vaccines, people with Other or no religious affiliation had relatively high rates of vaccine hesitancy, especially compared to Mainline Protestant and Catholic/Orthodox Christians. In the multivariable regression, which controlled for gender, age, region, education, race/ethnicity, and politics, those with no religious affiliation had significantly higher rates of COVID-19 non-vaccination. Previous research has indicated that religious beliefs could impact vaccination decisions across different nations [25,26,27]. For example, conservative Protestant communities have a significantly lower mean vaccine coverage than non-conservative Protestant communities in the Netherlands [25]. Parents who regularly attend religious services are more likely to decline HPV vaccines than parents who do not in the US [26]. Some possible explanations include the lack of trust in science, the belief in divine healing and protection from God [51,52], and conspiracy beliefs about vaccines. In addition, Catholic leaders in the US and Canada have raised concerns about some COVID-19 vaccines made using aborted fetal cells [53]. Of course, other moral and theological arguments could be made to support vaccination—for example, Pope Francis indicated that vaccination was a moral obligation [54]. The clergy may hold complex views on vaccination [55] but could nonetheless be important sources of information and education about vaccines.

In general, these important groups’ decision to receive vaccinations—Republicans, NH Black participants, and those with Other or no religious affiliation—is influenced more by historical and cultural factors than by their socioeconomic standing. In particular, denial and misinformation about the coronavirus on right-leaning media sources, racial disparities, distrust of the government, a lack of faith in science, a belief in divine healing and divine protection, and conspiracy theories about vaccines all contribute to the long-lasting roots of vaccine hesitancy and the non-receipt of vaccines among those populations. Thus, our future direction is to develop new interventions to target those demographic subgroups to reduce vaccine hesitancy and enhance vaccination rates. Since it can be difficult to change people’s attitudes toward vaccinations in general, efforts should emphasize lowering misinformation on social media via improving vaccine risk messages. Government, healthcare agencies, and trusted sources of information within these communities should tell the general public about vaccines in a reliable, evidence-based manner that is specific to people from various cultural backgrounds. Additionally, vaccination and communication campaigns are needed, which could facilitate culturally appropriate interpersonal communication about vaccines, and vaccination campaigns could address misinformation and conspiracy theories about vaccines. These campaigns could also help to increase confidence in vaccines and trust in the government.

### Limitations

The limitation of this study is related to the sampling method, since the study was conveniently sampled through the internet. Although we removed respondents who finished the survey in less than 180 s, there is still a chance that some respondents may have been included who were less focused and honest than others. People without internet access or who are from lower socioeconomic groups are less likely to be included in the study. Nonetheless, some evidence has been generated indicating that convenience samples can provide robust measurement estimates [56]. The final sample size has fewer participants compared to the pre-analysis populations after excluding those who took a short time. Based on the online survey using a convenience sample, we were unable to explore regional differences (i.e., state or municipality) because of the smaller sample size. Therefore, given the small sample size, additional data collected cross-sectionally or longitudinally that encompass bigger study populations with more diverse socioeconomic groups could be used in subsequent research to support the findings of this study. Additional questions, such as those about mistrust or experiences with vaccination, could inform future analyses of vaccine hesitancy development.

## 5. Conclusions

The study examined how general vaccination hesitancy and COVID-19 and flu vaccine non-receipt among US adults in August 2022 are impacted by race, religion, and political affiliation. In terms of prevalence, vaccine hesitancy was more prevalent among Republicans, those with no or an “Other” religious affiliation, and NH Black participants. The overlap between vaccine hesitancy and the refusal of the COVID-19 and flu vaccines was substantial but not exact. We discovered significant patterns in the general vaccine hesitancy and the non-receipt of the COVID-19 and flu vaccinations. Republicans, Non-Hispanic (NH) Black participants, and participants who identified as having no religion all had significantly higher rates of general vaccine hesitancy and the non-receipt of the COVID-19 vaccine. In a multivariable analysis controlling for demographic attributes, patterns of vaccine hesitancy and the non-receipt of COVID-19 vaccination did not vary, indicating a substantial overlap and potential spillover in vaccine hesitancy over the course of the pandemic. Because changing people’s opinions regarding vaccinations is generally a challenge, different interventions specific to demographic subgroups may be necessary.

## Figures and Tables

**Table 1 ijerph-20-03376-t001:** The prevalence of vaccine hesitancy in general, the non-receipt of COVID-19 vaccination, and the non-receipt of flu vaccination by demographic characteristics; August 2022.

		Count (Column %)	Vaccine Hesitancy Count (Row %)	Non-Receipt of COVID-19 Vaccination Count (Row %)	Non-Receipt of Flu Vaccination Count (Row %)
Total count		700 (100%)	319 (41%)	137 (17%)	288 (36%)
Gender			*p* = 0.1887	*p* = 0.0022	*p* = 0.2532
	Male	327 (48%)	133 (38%)	44 (12%)	116 (34%)
	Female	357 (52%)	163 (43%)	83 (22%)	167 (38%)
	Other	3 (0.2%)	0	0	0
Age			*p* = 0.0009	*p* = 0.1014	*p* = 0.0124
	18–24	181 (12%)	92 (54%)	44 (26%)	93 (48%)
	25–34	183 (19%)	85 (51%)	33 (18%)	78 (42%)
	35–44	197 (17%)	76 (40%)	30 (14%)	73 (36%)
	≥45	126 (52%)	43 (34%)	20 (16%)	39 (31%)
Region			*p* = 0.7764	*p* = 0.2980	*p* = 0.1933
	US West	142 (23%)	59 (36%)	21 (14%)	61 (33%)
	US Midwest	115 (22%)	53 (43%)	25 (18%)	46 (29%)
	US South	275 (36%)	119 (41%)	62 (21%)	120 (43%)
	US Northeast	155 (20%)	65 (43%)	19 (12%)	56 (35%)
Education			*p* = 0.0035	*p* < 0.0001	*p* = 0.0592
	≤High school	197 (24%)	100 (47%)	61 (26%)	97 (42%)
	Associate’s	177 (26%)	89 (51%)	39 (24%)	85 (43%)
	Bachelor’s	197 (28%)	66 (28%)	14 (4%)	66 (30%)
	Graduate	116 (21%)	41 (39%)	13 (15%)	35 (29%)
Race/Ethnicity			*p* = 0.0075	*p* < 0.0001	*p* = 0.0006
	Other	43 (7%)	18 (45%)	5 (11%)	23 (51%)
	NH Black	84 (11%)	47 (60%)	32 (39%)	51 (59%)
	Hispanic	113 (16%)	60 (48%)	16 (10%)	40 (32%)
	NH white	447 (65%)	171 (35%)	74 (16%)	169 (31%)
Religion			*p* = 0.0419	*p* < 0.0001	*p* < 0.0001
	Catholic /Orthodox	97 (19%)	40 (32%)	13 (11%)	37 (30%)
	Evangelical	170 (25%)	74 (46%)	23 (13%)	56 (32%)
	Mainline	43 (10%)	13 (27%)	7 (7%)	15 (22%)
	Other Christian	28 (3%)	12 (32%)	8 (20%)	15 (54%)
	Jewish	29 (5%)	13 (52%)	2 (10%)	5 (21%)
	Muslim	39 (4%)	11 (34%)	0 (0%)	4 (7%)
	Other	60 (6%)	42 (64%)	21 (33%)	34 (61%)
	Nothing	221 (29%)	91 (42%)	53 (28%)	117 (47%)
Political Affiliation			*p* = 0.0414	*p* = 0.0029	*p* = 0.0004
	Democrat	307 (40%)	102 (33%)	35 (11%)	105 (32%)
	Independent	186 (30%)	110 (45%)	62 (27%)	118 (51%)
	Republican	184 (30%)	84 (47%)	30 (16%)	60 (27%)

Note: NH, non-Hispanic.

**Table 2 ijerph-20-03376-t002:** The demographic characteristics of the non-receipt of COVID-19 and flu vaccinations; August 2022.

		Received Both COVID-19 and Flu Vaccination	Received Only COVID-19 Vaccination	Received Only Flu Vaccination	Received Neither	*p*-Value
Total count		488 (61%)	176 (22%)	25 (3%)	112 (14%)	
Gender	Male	212 (65%)	79 (23%)	7 (2%)	37 (11%)	0.0353
	Female	180 (57%)	99 (21%)	15 (4%)	68 (17%)	
	Other	3 (100%)	0	0	0	
Age	18–24	84 (46%)	57 (27%)	8 (5%)	36 (21%)	0.0843
	25–34	105 (56%)	49 (26%)	4 (2%)	29 (16%)	
	35–44	121 (61%)	49 (25%)	6 (3%)	24 (11%)	
	≥45	85 (66%)	23 (18%)	4 (3%)	16 (13%)	
Region	US West	80 (65%)	42 (21%)	2 (2%)	19 (12%)	0.4638
	US Midwest	68 (65%)	26 (16%)	5 (5%)	20 (13%)	
	US South	148 (54%)	70 (25%)	12 (3%)	50 (18%)	
	US Northeast	99 (64%)	40 (24%)	3 (2%)	16 (11%)	
Education	≤High school	96 (54%)	44 (19%)	8 (4%)	53 (23%)	0.0005
	Associate’s	87 (52%)	53 (24%)	7 (5%)	32 (19%)	
	Bachelor’s	135 (70%)	54 (26%)	2 (<0.5%)	12 (4%)	
	Graduate	77 (68%)	27 (17%)	5 (4%)	8 (11%)	
Race/Ethnicity	Other	22 (49%)	18 (40%)	0	5 (11%)	<0.0001
	NH Black	30 (36%)	24 (25%)	5 (5%)	27 (35%)	
	Hispanic	72 (66%)	28 (25%)	4 (2%)	12 (7%)	
	NH white	271 (65%)	108 (19%)	13 (3%)	61 (12%)	
Religion	Catholic /Orthodox	59 (69%)	27 (20%)	3 (1%)	10 (10%)	<0.0001
	Evangelical	115 (66%)	36 (21%)	3 (2%)	20 (11%)	
	Mainline	27 (76%)	10 (17%)	2 (2%)	5 (5%)	
	Other Christian	11 (40%)	9 (39%)	2 (5%)	6 (15%)	
	Jewish	24 (79%)	3 (11%)	0	2 (10%)	
	Muslim	39 (93%)	4 (7%)	0	0	
	Other	24 (37%)	15 (30%)	2 (2%)	19 (31%)	
	Nothing	96 (46%)	74 (26%)	10 (7%)	43 (21%)	
Political Affiliation	Democrat	200 (65%)	81 (24%)	11 (3%)	24 (7%)	0.0004
	Independent	72 (47%)	63 (27%)	7 (3%)	55 (24%)	
	Republican	123 (70%)	34 (14%)	4 (3%)	26 (13%)	

Note: NH, non-Hispanic.

**Table 3 ijerph-20-03376-t003:** Patterns of vaccine hesitancy and the non-receipt of COVID-19 vaccines among 700 US adult participants; August 2022.

		Vaccine Hesitancy OR (95% CI)	Non-Receipt of COVID-19 Vaccination OR (95% CI)	*p*-Value
Gender	Male	REF	REF	0.1325
	Not Male	1.3 (0.8, 2.0)	2.2 (1.2, 4.1)	
Age	18–24	0.9 (0.5, 1.5)	0.9 (0.4, 1.8)	0.7359
	25–34	REF	REF	
	35–44	0.7 (0.4, 1.2)	0.9 (0.5, 1.9)	
	≥45	0.6 (0.3, 1.1)	1.0 (0.5, 2.0)	
Region	US West	0.6 (0.3, 1.2)	0.9 (0.3, 2.3)	0.4712
	US Midwest	1.1 (0.5, 2.2)	1.4 (0.6, 3.7)	
	US South	0.8 (0.4, 1.4)	1.6 (0.7, 3.8)	
	US Northeast	REF	REF	
Education	≤High school	1.6 (0.9, 3.0)	6.2 (2.7, 14.6)	0.0938
	Associate’s	2.6 (1.4, 4.9)	5.7 (2.5, 13.1)	
	Bachelor’s	REF	REF	
	Graduate	1.8 (0.9, 3.6)	4.0 (1.5, 10.8)	
Race/Ethnicity	Other	1.5 (0.7, 3.6)	0.7 (0.2, 2.7)	0.0723
	NH Black	3.0 (1.5, 6.0)	3.7 (1.8, 7.8)	
	Hispanic	1.6 (0.9, 2.9)	0.6 (0.3, 1.1)	
	NH white	REF	REF	
Religion	Catholic/Orthodox	REF	REF	0.0697
	Evangelical	1.7 (0.8, 3.4)	1.1 (0.4, 3.2)	
	Other	2.8 (1.3, 6.1)	1.4 (0.5, 4.1)	
	Other Christian	0.9 (0.3, 2.2)	0.6 (0.2, 1.7)	
	Nothing	1.4 (0.7, 2.7)	2.6 (1.0, 6.8)	
Political Affiliation	Democrat	REF	REF	0.2447
	Independent	2.0 (1.2, 3.4)	3.9 (2.1, 7.3)	
	Republican	2.3 (1.3, 4.1)	2.6 (1.2, 5.5)	

Note: NH, non-Hispanic.

## Data Availability

The dataset is available at: https://doi.org/10.6084/m9.figshare.21797729.

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
