# Peer review of "Spillover of Vaccine Hesitancy into Adult COVID-19 and Influenza: The Role of Race, Religion, and Political Affiliation in the United States"

_ijerph, 2023, doi:10.3390/ijerph20043376_

Round 1
Reviewer 1 Report
Thank you for the possibility to review the manuscript titled: “Spillover of vaccine hesitancy into adult COVID-19 and influenza: the role of race, religion, and political affiliation in the United States”. The manuscript is interesting, controversial, brings up several important points for analysis in the future. There are several minor corrections:
-“Gender” is a psychological term. I believe that authors meant sex, which a biological analogue.
-Table 1 is positioned in the material and methods section. I think that this table already represents the initial results of the study and can be moved to the results section.
-The study brings up several “key” groups that have lower trust for vaccination. I believe that the decision of these groups are based on historical and cultural reasons more than their socio-economical position. This point deserves more analysis in the discussion section.
-Please include a short section in the discussion part of the manuscript about possible future directions for research.
Please take into account the recommendations in the spirit of improving the quality of the submission.
Author Response
Reviewer 1
Thank you for the possibility to review the manuscript titled: “Spillover of vaccine hesitancy into adult COVID-19 and influenza: the role of race, religion, and political affiliation in the United States”. The manuscript is interesting, controversial, brings up several important points for analysis in the future. There are several minor corrections:
-“Gender” is a psychological term. I believe that authors meant sex, which a biological analogue.
Authors’ response: We consider people with “other” gender identity, so we use gender instead of sex on the table. We include this reference (line 152): “We asked people about their gender identity using the categories Female, Male, and Other suggested by the American Association of Public Opinion Researchers [40].”
-Table 1 is positioned in the material and methods section. I think that this table already represents the initial results of the study and can be moved to the results section.
Authors’ response: Thank you for the suggestion. We have move table 1 to the result section.
-The study brings up several “key” groups that have lower trust for vaccination. I believe that the decision of these groups are based on historical and cultural reasons more than their socio-economical position. This point deserves more analysis in the discussion section.
Authors’ response:
We added
“In general, these important groups' decisions to receive vaccinations—Republicans, NH Black participants, and those with Other or no religious affiliation—are influenced more by historical and cultural factors than by their socioeconomic standing. In particular, denial and misinformation about the coronavirus on right-leaning media sources, racial disparities, distrust of the government, a lack of faith in science, a belief in divine healing, divine protection and conspiracy theories about vaccines all contribute to the long-lasting roots of vaccine hesitancy and non-receipts of vaccines in those populations.”
to line 436. We also included some of the historical and cultural reasons in the introduction.
-Please include a short section in the discussion part of the manuscript about possible future directions for research.
Authors’ response:
We added “Thus, our future direction is to develop new interventions to target those demographic subgroups to reduce vaccine hesitancy and enhance vaccination rates. Since it can be difficult to change people's attitudes toward vaccinations in general, efforts should emphasize lowering misinformation on social media via improving vaccine risk messages. Government and healthcare agencies should tell the general public about vaccines in a reliable, evidence-based manner that is specific to people from various cultural backgrounds. Additionally, vaccination and communication campaigns are needed, which communication campaigns could facilitate culturally appropriate interpersonal communication about vaccines, and vaccination campaigns could address misinformation and conspiracy theories about vaccines. These campaigns could also help to increase confidence in vaccines and trust in the government.” in the end of discussion.
Please take into account the recommendations in the spirit of improving the quality of the submission.
Authors’ response: We appreciate all the comments from the editor and reviewers and believe our manuscript has been greatly strengthened.
Reviewer 2 Report
Zhang et al. showed that race, religion, and political affiliations are key demographic factors that affect vaccine hesitancy, and non-receipt of COVID-19 and influenza vaccines. The authors used an online sampling in which over 700 American adults drawn from different locations were recruited. Their results show that race, religion, and political affiliations influence vaccine hesitancy and non-receipt. However, I do have the following concerns.
Major concerns
1: What is the scientific basis for eliminating participants that answered in less than 180 seconds?
2: Is 700 participants actually an accurate representation of over 320 million people? I think more participants should have been recruited given that it was an online survey
3: Does any question in the survey ask about vaccine mistrust? I think any respondent with any family member or friend that has had a bad experience with vaccines will be hesitant.
4: The paper aimed at addressing three key questions, but the conclusion part of the abstract listed only the third aim. Any reasons for this?
Minor concerns
1: There should be a citation on line 64 after adult vaccine hesitancy
2: Lines 78-83 should be rephrased and maybe reported as a percentage decrease from the previous year (Example In 2022, only 44% of nurses in HK received flu vaccine compared to the previous year) so it will be easier to understand.
3: Citation on line 95 after the setting of the epidemic.
4: In lines, 107/109, should be political affiliations? I do not understand what political affliction means.
5: I think the title should be edited to reflect the questions the paper addressed.
Author Response
Reviewer 2
Zhang et al. showed that race, religion, and political affiliations are key demographic factors that affect vaccine hesitancy, and non-receipt of COVID-19 and influenza vaccines. The authors used an online sampling in which over 700 American adults drawn from different locations were recruited. Their results show that race, religion, and political affiliations influence vaccine hesitancy and non-receipt. However, I do have the following concerns.
Major concerns
1: What is the scientific basis for eliminating participants that answered in less than 180 seconds?
Authors’ response: We include this info now (line 118):
“We omitted individuals who took less than 180 seconds to complete the survey, which we judged to be the shortest time to thoughtfully consider each question, based on extensive pretesting from multiple individuals.”
--
2: Is 700 participants actually an accurate representation of over 320 million people? I think more participants should have been recruited given that it was an online survey
Authors’ response: We now include this sample size calculation (line 122): “Our sample size calculation was based on the overall project's goal to precisely estimate the proportion of people who were vaccinated at a particular period, with a margin of error of 4%, an alpha of 0.05, and a power of 80% [37]. A sample size of 800 would be required to estimate an outcome proportion of 50% (a statistically conservative estimate of the target outcome).
“
We also provide more context in the limitations (line 373): “The limitation of this study is related to the sampling method since the study was conveniently sampled through the internet. Although we removed respondents who finished the survey in less than 180 seconds, there is still a chance that some respondents may have been included who were less focused and honest than others. Besides, people without internet access or from lower socioeconomic groups are less likely to be included in the study. Nonetheless, some evidence has been generated that convenience samples can provide robust measurement estimates [56].”
--
3: Does any question in the survey ask about vaccine mistrust? I think any respondent with any family member or friend that has had a bad experience with vaccines will be hesitant.
Authors’ response: We agree this is an interesting point, but it was, unfortunately, not asked in this survey. We include the following now in the limitations (line 385):
“Additional questions, like those about mistrust or experiences with vaccination, could inform future analyses of vaccine hesitancy development.”
--
4: The paper aimed at addressing three key questions, but the conclusion part of the abstract listed only the third aim. Any reasons for this?
Authors’ response: Unfortunately due to abstract space limitations, we condensed our discussion of the aims. But our results in the abstract do include some of this information anyway (line 17): “Of the 700 participants, 49% of the respondents were classified as having general vaccine hesitancy, 17% had not received the COVID-19 vaccine, and 36% had not had flu vaccinations. In the multivariable analysis, general vaccine hesitancy and non-receipt of COVID-19 vaccines were significantly higher in Non-Hispanic Black participants, those with no religious affiliation, and Republicans and Independents. “
--
Minor concerns
1: There should be a citation on line 64 after adult vaccine hesitancy
Authors’ response:
We have added in citations to this sentence and revamped the introduction to provide more context for the study.
--
2: Lines 78-83 should be rephrased and maybe reported as a percentage decrease from the previous year (Example In 2022, only 44% of nurses in HK received flu vaccine compared to the previous year) so it will be easier to understand.
Authors’ response:
We simplified this sentence to be (line 66): “As another example, influenza vaccine uptake in different areas is suboptimal. Researchers found declines in influenza vaccination coverage among Hong Kong nurses in 2017-2018, with 79% of them expressing moderate or greater vaccine hesitancy [22].”
--
3: Citation on line 95 after the setting of the epidemic.
Authors’ response:
Citations have been provided
--
4: In lines, 107/109, should be political affiliations? I do not understand what political affliction means.
Authors’ response: Thank you for this correction, it should be affiliation.
--
5: I think the title should be edited to reflect the questions the paper addressed.
Authors’ response: We appreciate the comment and argue it should stay as is. But we have edited the paper to be more consistent with the title. For instance, in the discussion we now write: (line 277)
“A large focus of this analysis was to discover if demographic patterns varied between vaccine hesitancy, COVID-19 vaccine non-receipt, and flu vaccine non-receipt. We did not find any substantial significant difference in patterns of vaccine hesitancy and COVID-19 non-vaccination. This finding also speaks to the strong possibility of continued spillover in attitudes across vaccines. A definition of the WHO SAGE Working Group on Vaccine Hesitancy stresses that vaccine hesitancy is type specific, i.e., could vary across vaccines for different pathogens [8]. However our study, although finding that the prevalence of uptake for various vaccines may vary, the patterns of uptake or vaccine hesitancy are largely the same between different sociodemographic groups. As a result, surveys of vaccine hesitancy in general could identify hot spots of non-vaccination for this outbreak and others. And additionally, these trends speak to the difficulty in increasing the uptake of COVID-19 vaccination, as changing minds about vaccines in general is relatively difficult [41], and the interventions to do so would be different than increasing access or convenience of vaccination. “
Reviewer 3 Report
The pandemic has brought with it great medical and scientific challenges. Given the impact on the world's health systems and the battle against time that the manufacture and distribution of vaccines represented, together with the constant appearance of viral variants, the increase in cases and associated deaths, the need arises for this type of studies that seek to reveal the factors that limit the population to be vaccinated, such as age, sex, politics and religion
It seems to me an interesting and well presented work, my suggestions would be: to improve the wording of the summary and the introduction, the age could be stratified in less wide intervals at the moment of the data analysis. It is also important to mention the high content through the media and social networks, some of them lacking scientific qualities that have a direct or indirect impact on the behavior and beliefs of people, and that could condition that they do not want to be vaccinated.
Author Response
Reviewer 3
The pandemic has brought with it great medical and scientific challenges. Given the impact on the world's health systems and the battle against time that the manufacture and distribution of vaccines represented, together with the constant appearance of viral variants, the increase in cases and associated deaths, the need arises for this type of studies that seek to reveal the factors that limit the population to be vaccinated, such as age, sex, politics and religion
It seems to me an interesting and well presented work, my suggestions would be: to improve the wording of the summary and the introduction, the age could be stratified in less wide intervals at the moment of the data analysis. It is also important to mention the high content through the media and social networks, some of them lacking scientific qualities that have a direct or indirect impact on the behavior and beliefs of people, and that could condition that they do not want to be vaccinated.
Authors’ response: We appreciate the comments and have extensively edited the introduction to better reflect the theoretical underpinning of the research.
We have now edited the discussion to emphasize news media (line 312):
“During the start of the pandemic, denial and misinformation about the coronavirus erupted on right-leaning media sources. For instance, Fox News, more popular among Republican voters, was much more likely than CNN or MSNBC to use a skeptical narrative that minimized the dangers posed by the COVID-19 outbreak, including words like “normal flu” and “political weapon [46].” Only 38% of Fox News viewers expressed concern about the coronavirus in a March 2020 survey, compared to 71% of CNN viewers [47].”
–
Reviewer 4 Report
The data in Table 1 and Table 2 do not conform to statistical specifications ,and the datas are unreliable.
Author Response
Reviewer 4
The data in Table 1 and Table 2 do not conform to statistical specifications ,and the datas are unreliable.
Authors’ response: We appreciate the comment. We now highlight in the methods that (line 163): “A chi-square test is used to determine the p-value, except for variables with low cell counts, where Fisher’s exact test was used to adjust for a small sample size.”
And we describe in the methods in more detail how different variables were measured, including past measurements of reliability:
Line 131:
“We used the adult Vaccine Hesitancy Scale (aVHS) to assess vaccine hesitancy [38]. Briefly, this is a 10-item scale, and the sum of each question’s responses is dichotomized at 25 to partition individuals into being hesitant or not. The face validity, concurrent validity, and reliability of this question are reported elsewhere [38].
For COVID-19 vaccination, participants were asked at the start of the questionnaire: “Have you personally received at least one dose of a Covid-19 vaccine or not?” Those who answered “No” are categorized as having not received the COVID-19 vaccine. Vaccine questions were based on survey items from the Kaiser Family Foundation COVID-19 Vaccine Monitor.
The non-receipt of the flu vaccine was obtained by asking participants: “Have you received a flu shot for the current flu season – that began October 2021, or not?” Those who answered “No” are categorized as non-receipt of the flu vaccine.”
Round 2
Reviewer 2 Report
The authors sufficiently addressed the concerns I raised except for
Does any question in the survey ask about vaccine mistrust? I think any respondent with any family member or friend that has had a bad experience with vaccines will be hesitant
Author Response
Does any question in the survey ask about vaccine mistrust? I think any respondent with any family member or friend that has had a bad experience with vaccines will be hesitant.
Authors' response: Our apologies for lack of a previous response. We unfortunately did not collect that data, although we agree it would be interesting.
Reviewer 4 Report
Accept
Author Response
Thank you for your thorough review of our paper!